# Integrated microbiology and metabolomics analysis reveal the fermentation process and the flavor development in cigar tobacco leaf

Guanghai Zhang,[1] Yue He,[2] Wanlong Yang,[3] Ziyi Liu,[4] Zhonglong Lin,[2] Tikun Zhang,[4] Xiaohui He,[3] Huachan Xia,[1] Mengting Huo,[3] Heng Yao,[1] Gaokun Zhao,[1] Yuping Wu,[1] Guanghui Kong[1]

**ABSTRACT**   The purpose of this study was to elucidate how the microbiota affects the metabolic state and characteristic flavor development of cigar tobacco leaves (CTL) in the fermentation process through microbial metabolism and co-metabolism with the host. The results showed that core bacterial communities in the fermentation process were *Cyanobacteria*, *Pseudomonas*, and *Staphylococcus*, and *Aspergillus*, *Penicillium*, and *Inocybe* were the most metabolically active fungi. *Pseudomonas fulva*, *Staphylococcus nepalensis*, *Bacillus subtilis*, *Stenotrophomonas rhizophila*, *Alternaria alternata*, and *Aspergillus cristatus* could degrade carbon and nitrogen compounds, such as protein, starch, lignin, and nicotine. There were 12 common non-volatile metabolites and three common volatile metabolites in DH, LC, PE, and YX before and after fermentation, of which menatetrenone, solanidine, γ-glutamylphenylalanine, carotol, and phenol, 4-(1,1,3,3-tetramethylbutyl) were significantly different before and after fermentation. The synthesis and degradation metabolism of various amino acids, alkaloids, and flavonoids are the key metabolic pathways of characteristic flavor development during CTL fermentation. Co-occurrence and interaction patterns showed that seven bacteria and 12 fungi were strictly linearly positively or negatively correlated with 72 and 55 volatile compounds, respectively. In conclusion, this study preliminarily confirmed that CTL fermentation is a microbially mediated carbon–nitrogen coupling metabolism. The carbohydrates in tobacco leaves were largely decomposed and consumed, providing energy sources for microorganisms and the carbon skeleton required for cell construction. The nitrogen-containing macromolecular compounds were degraded to form volatile compounds or flavor precursors with typical flavor.

**IMPORTANCE**   The development of the metabolic state and characteristic flavor of cigar tobacco during fermentation is the key to process control. Innovative discoveries in the development of the metabolic state and characteristic flavor of cigar tobacco during fermentation are the key to process control. Innovative discoveries of core functional microorganisms and key metabolites were made during fermentation, suggesting potential pathways for carbon and nitrogen metabolisms. We demonstrate for the first time that cigar tobacco leaf fermentation is a microbially mediated carbon–nitrogen coupling metabolism. Many carbohydrates in tobacco leaves are decomposed and consumed to provide energy source and carbon skeleton for the construction of microbial cells, and nitrogen-containing macromolecular compounds are degraded to form volatile compounds or flavor precursors with typical flavors.

**KEYWORDS**   cigar tobacco leaf, solid state fermentation, metabolomics, microbiome

**Peer Reviewers** Liwei Hu, Zhengzhou Tobacco Research Institute of CNTC, Zhengzhou, China; Qin Liu, The Chinese University of Hong Kong, Hong Kong, China

Address correspondence to Guanghui Kong, 13908776036@163.com.

The authors declare no conflict of interest.

Cigar tobacco leaf stacking fermentation involves enzymatic actions of many microorganisms and is an efficient and environmentally benign process for degrading macromolecular organic compounds (1). Various enzymes within cells are the main catalytic factors for various metabolic pathways during cigar tobacco leaf (CTL) processing, promoting the transformation of tobacco chemical substances. Microorganisms play an important role in the fermentation process (2, 3). During growth and reproduction, microorganisms use organic compounds in tobacco leaves as substrates to degrade and convert macromolecular substances into amino acids, alcohols, aldehydes and ketones, organic acids, and other heterocyclic aromatic substances (4, 5). However, the coupling of material conversion and microbial community succession during CTL fermentation has not been clarified. The interaction between substance transformation, metabolic pathways, and microorganisms during CTL fermentation is not yet clear. Up to now, the mechanism of quality formation during the fermentation process of cigar tobacco leaves is still a focus and difficulty of research.

High-throughput techniques and other omics techniques, such as metagenomics and metabolomics, have advanced rapidly, and, as a result, there is increasing recognition of the importance of integrating these approaches. Compared to single-omics ones, multi-omics analyses more accurately delineate microbial and molecular characteristics, so they are more apt to elucidate the mechanism of fermentation (6). Headspace solid-phase microextraction–gas chromatography–mass spectrometry (HS-SPME–GC–MS) and ultraperformance liquid chromatograph–electrospray ionization-tandem mass spectrometer (UPLC-ESI–MS/MS) have been widely used in the analysis of substance metabolism and flavor formation in the fermentation process of soybean food (7), Chinese liquor (8), or tea (9). However, the application of flavor development and formation in cigar tobacco leaf fermentation has rarely been reported.

Therefore, this study investigated the microbial community and metabolic function during solid-state fermentation of cigar tobacco from different origins and analyzed whether the microorganisms were closely related to metabolite synthesis. Illumina NovaSeq sequencing of 16S rRNA and ITS genes was performed. The tobacco leaves before and after fermentation were analyzed by HS-SPME–GC–MS and ESI–MS/MS. The purpose of this study was to investigate the response of microbial diversity, community structure, and metabolic function in CTL fermentation and provide a basis for determining microbial groups involved in macromolecular substance transformation and metabolism and guiding the optimization and adjustment of the CTL fermentation technology.

## RESULTS

### Bacteria succession before and after fermentation

A total of 1,942,260 bacterial 16S rRNA raw reads were obtained from 24 samples by amplicon sequencing, and the effective clean data of 1,876,963 bacteria from 96.69% raw reads were used for the subsequent analysis (Table S1). Based on the 97% sequence similarity principle, operational taxonomic units (OTU) were used for species annotation and denoise. Among all the amplicon sequence variants (ASVs) identified in this study, the bacteria shared by all four locations had 27 ASVs. The PEF0 vs PEF5, LCF0 vs LCF5, DHF0 vs DHF5, and YXF0 vs YXF5 had 29 vs 1,852, 279 vs 69, 41 vs 38, and 134 vs 898 unique ASVs, respectively (Fig. 2A). The Chao1 and Shannon index suggested the highest bacterial diversity was in the PE and YX samples. In the overall statistical analysis, PEF5 had a higher bacterial Shannon than PFF0 ($P < 0.05$) (Fig. 1B). The β-diversity of bacteria and α-diversity were consistent (Fig. 2C).

At the genus level, the core bacterial communities in the fermentation process were *Cyanobacteria*, *Pseudomonas*, and *Staphylococcus* (Fig. 2D), while the relative abundance of some other unknown bacteria accounts for a large proportion. To further analyze the major different bacterial communities of pre- and post-fermentation tobacco leaves in the four origins, the differences between groups were analyzed by Metastats. The results showed that DHF0 vs. DHF5, LCF0 vs. LCF5, PEF0 vs. PEF5, and YXF0 vs. YXF5 had three,

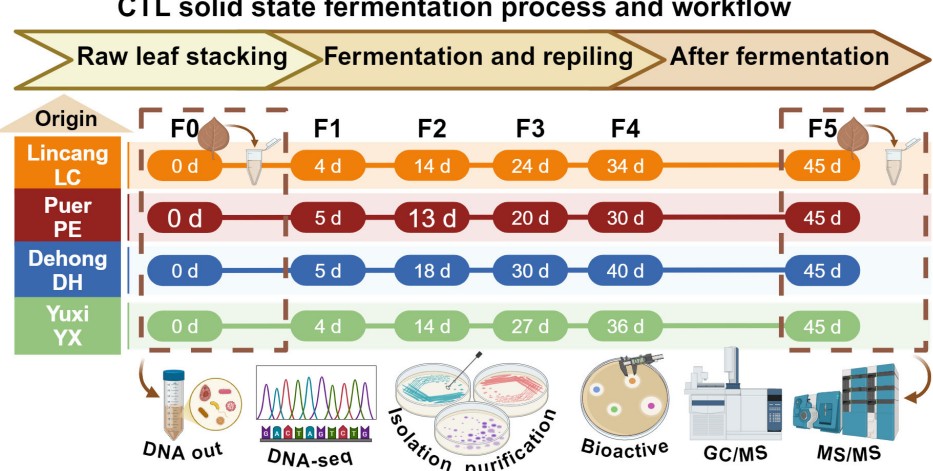

**FIG 1** Overview of the CTL fermentation experiments and sample collection.

six, eight, and seven significantly different bacteria (top 35), respectively. The *Corynebacterium*, *Massilia*, and *Pseudomonas* before and after fermentation of CTL from DH origin are the main bacteria. The bacteria with significant differences in CTL from the LC region were *Cyanobacteria*, *Staphylococcus*, *Pseudomonas*, *Proteus*, *Paracoccus*, *Pantoea*, *Kosakonia*, *Anaerostipes*, and *Acinetobacter*. *Acinetobacter*, *Staphylococcus*, *Kosakonia*, *Pantoea*, *Cyanobacteria*, and *Pseudomonas* were the main difference in bacteria before and after fermentation of CTL from the PE region. CTL from the YX origin have significant differences in *Staphylococcus*, *Aerococcus*, *Clostridiaceae*, *Corynebacterium*, *Alloprevotella*, *Pantoea*, and *Parasutterella* (Fig. 2E).

## Fungi succession before and after fermentation

A total of 1,984,844 fungal ITS raw reads were obtained from 24 samples by amplicon sequencing, and the effective clean data of 1,937,876 fungi from 97.58% raw reads were used for the subsequent analysis (Table S1). Based on the 97% sequence similarity principle, OTUs were used for species annotation and denoising. Among all the ASVs identified in this study, the fungi shared by all four locations had 20 ASVs. The PEF0 vs PEF5, LCF0 vs LCF5, DHF0 vs DHF5, and YXF0 vs YXF5 had 233 vs 174, 279 vs 133, 0 vs 134, and 0 vs 188 unique ASVs, respectively (Fig. 3A). The Chao1 and Shannon indices showed that there was no significant difference in the fungal diversity of PE, LC, DH, and YX samples. In the overall statistical analysis, the fungal Shannon index of YXF5 was significantly higher than that of YXF0 ($P < 0.05$), and there were no significant differences between PE, DH, and LC before and after fermentation (Fig. 3B). PE had the lowest β-diversity before and after fermentation, followed by LC, YX, and DH from the lowest to the highest (Fig. 3C).

The relative abundance of other core fungi (top 10) varied greatly among the four locations. Moreover, there was a clear succession pattern before and after fermentation. For example, the relative abundance of *Penicillium* and *Sampaiozyma* decreased after fermentation in PE regions, whereas that of *Trichomonascus* decreased (Fig. 3D). Metastats was used to further analyze the core fungal differences between before and after fermentation of CTL. The results showed that the significantly different fungi in DHF0 vs. DHF5 were *Sebacina*, *Aspergillus*, *Ascomycota*, *Microascus*, and *Inocybe*. *Staphylotrichum*, *Colletotrichum*, *Aspergillus*, *Sedis*, *Ascomycota*, *Didymella*, and *Inocybe* in LCF0 vs. LCF5 were the main significantly different fungi. The fungi with significant difference PEF0 vs. PEF5 in CTL were *Aspergillaceae*, *Fungi_gen_Incertae_sedis*, *Penicillium*, *Sampaiozyma*, *Microascus*, and *Trichomonascus*. The significantly different fungi in YXF0 vs. YXF5 were *Wallemia*, *Nigrospora*, *Colletotrichum*, *Botryosphaeria*, *Diaporthe*, *Aspergillus*,

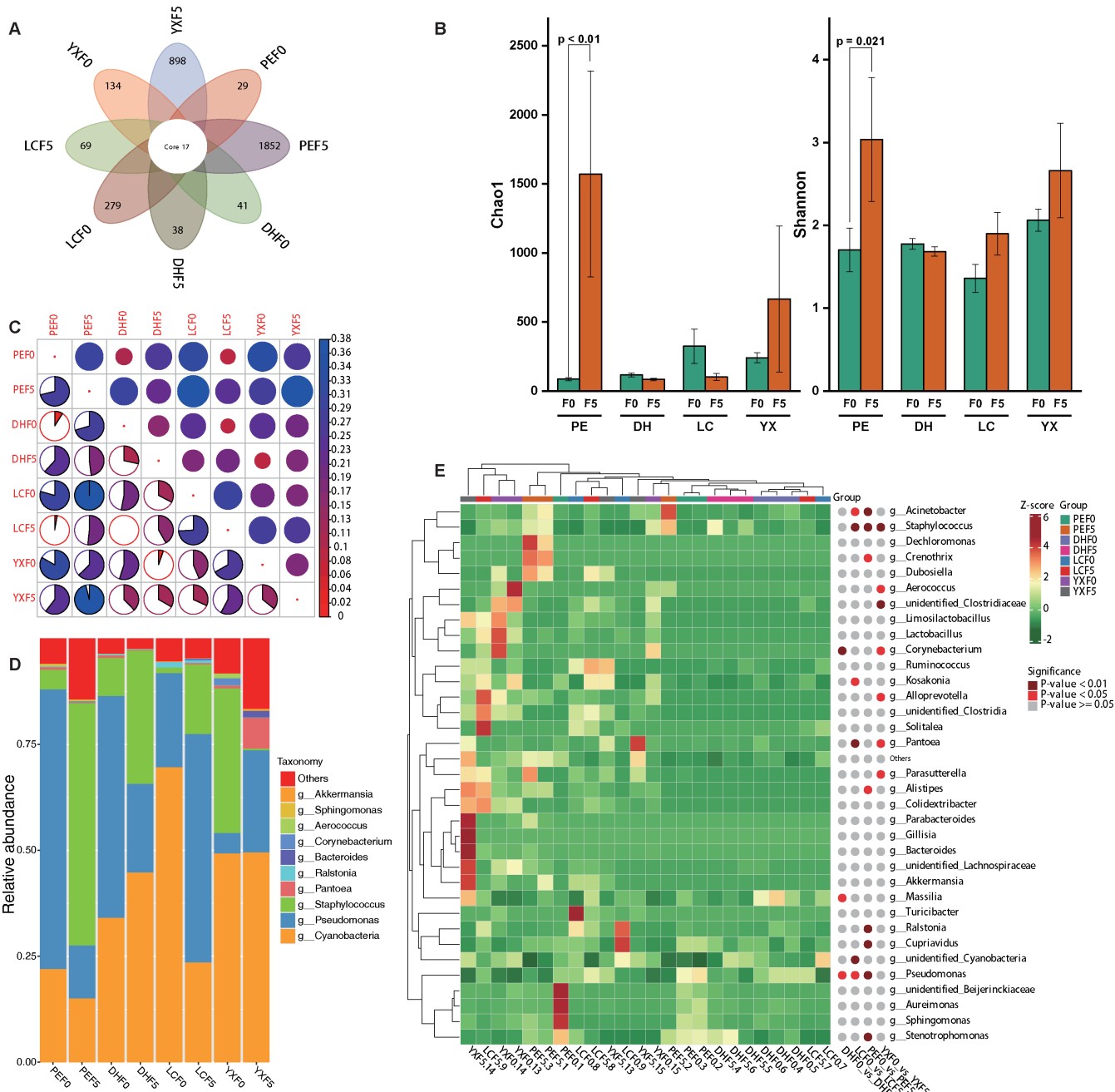

**FIG 2** Dynamic of bacterial communities before and after fermentation. (A) Venn diagram shows specific ASVs and shared ASVs of bacteria; (B) α-diversity of bacteria; (C) β-diversity of bacteria (the smaller the circle and the redder the color, the smaller the beta diversity value between samples); (D) Relative abundance of dominant bacteria (top 10); and (E) quantitative data heatmap based on ASV and significance difference identification by Metastats (top 35).

*Aspergillaceae*, *Penicillium*, *Sampaiozyma*, *Microascus*, *Cercospora*, and *Inocybe* (Fig. 2E). At the genus level, the core fungi in the fermentation process were *Aspergillus*, *Penicillium*, and *Inocybe*.

## Microbial biodegradability

In this study, before and after fermentation, CTL samples were selected for micro-bial isolation and purification. A total of 31 culturable bacteria were isolated from CTL samples before and after fermentation. They were mainly *Bacillus*, *Pseudomonas*,

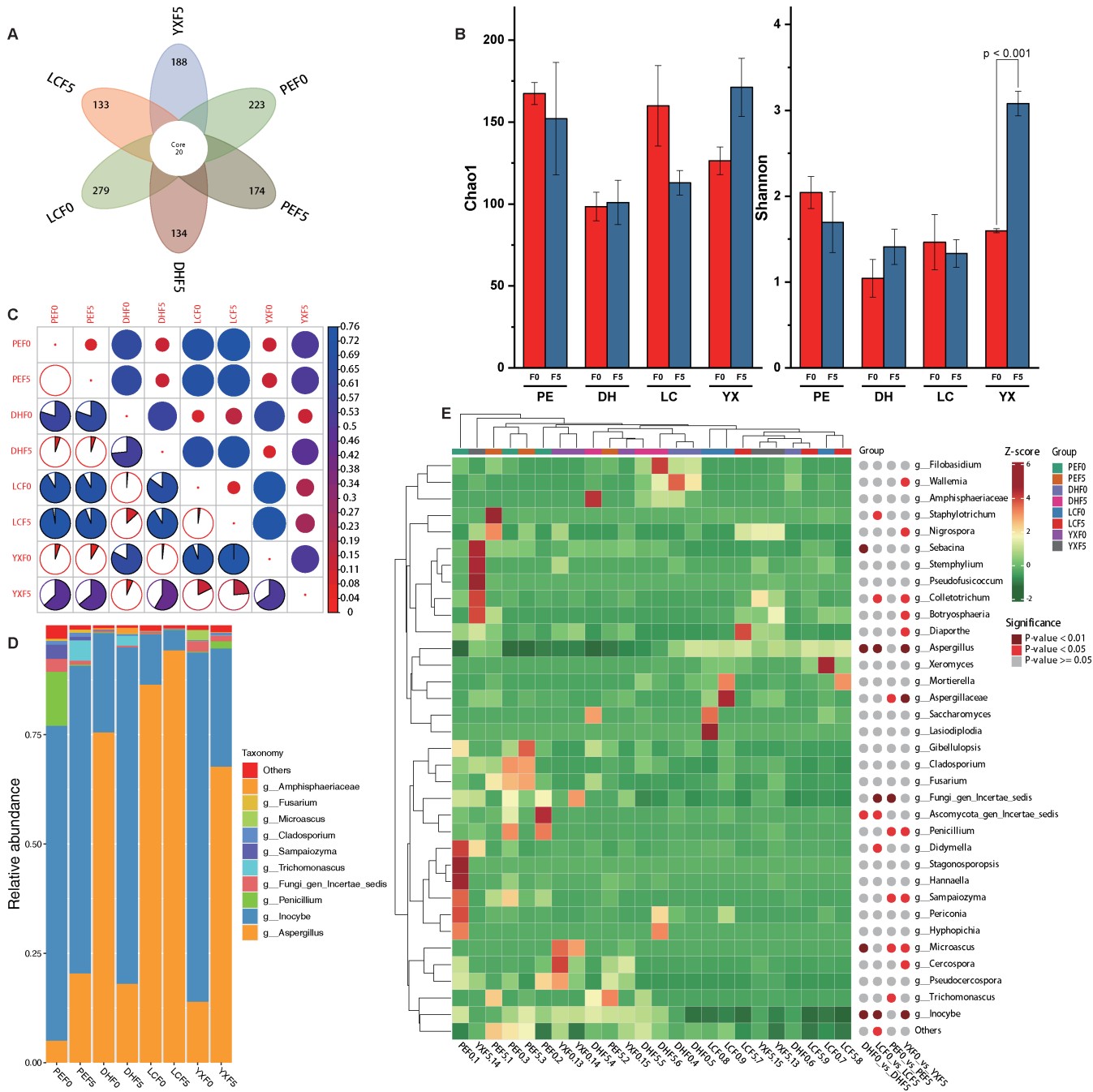

**FIG 3** Dynamic of fungal communities before and after fermentation. (A) Venn diagram shows specific ASVs and shared ASVs of fungi; (B) α-diversity of fungi; (C) β-diversity of fungi (the smaller the circle and the redder the color, the smaller the beta diversity value between samples); (D) relative abundance of dominant fungi (top 10); and (E) quantitative data heatmap based on ASV and significance difference identification by Metastats (top 35).

*Stenotrophomonas*, *Staphylococcus*, and *Arthrobacter*, and a total of 12 kinds of culturable fungi were isolated, mainly *Aspergillus*, *Alternaria*, *Arcopilus*, and *Rhizopus* (Table S2).

According to the OTU sequence of CTL samples and the molecular identification results of single strains, we selected four bacteria and two fungi with sequence similarity greater than 99% to analyze their ability to degrade macromolecular substances, such as protein, starch, lignin, and nicotine (Table 1; Fig. S1). For bacteria, *B. subtilis* had the ability to degrade all four substances; *P. fulva* had the ability to degrade three substances; *S. rhizophila* had a strong ability to degrade protein; and *S. nepalensis* had the ability to

**TABLE 1** Degradation of compounds by culturable bacteria and fungi

| Microbes | ID | Species | Protein | Starch | Lignin | Nicotine |
|---|---|---|---|---|---|---|
| Bacteria | C26-6 | *Bacillus subtilis* | 3.50 | 1.02 | 0.55 | 5.20 |
| | C26-7 | *Pseudomonas fulva* | 2.86 | /[a] | 1.89 | 1.72 |
| | F16-12 | *Stenotrophomonas rhizophila* | 8.83 | / | / | / |
| | F36-14 | *Staphylococcus nepalensis* | 4.56 | / | / | 5.32 |
| Fungi | F16-1 | *Alternaria alternata* | / | 0.97 | 0.42 | 4.56 |
| | F16-2 | *Aspergillus cristatus* | / | 0.42 | 0.43 | / |

[a]/ indicates that the microorganism has no degrading activity for the compound.

degrade protein and nicotine. *A. alternata* had the best degradation ability to nicotine but weak degradation ability to starch and lignin. *A. cristatus* has certain degradation ability for both starch and lignin, but the degradation ability is relatively weak.

## Metabolic profile before and after fermentation

A total of 531 primary metabolites, 560 secondary metabolites, and 14 other metabolites were obtained. The main substances in the primary metabolites were amino acids and derivatives, lipids, and organic acids, accounting for 72.88%. Phenolic acids, alkaloids, and flavonoids accounted for 88.21% of the secondary metabolites (Fig. 4A; Table S3). A total of 613 volatile metabolites were detected from 24 samples, including 138 terpenes, 90 lipids, 87 heterocyclic compounds, 61 ketones, and 55 hydrocarbons, accounting for 70% of all volatile metabolites (Fig. 4B; Table S3).

The differential metabolites of CTL before and after fermentation in four producing areas were further analyzed. There were 123, 255, 199, and 422 significantly different non-volatile metabolites in DHF0 vs DHF5, LCF0 vs LCF5, PEF0 vs PEF5, and YXF0 vs YXF5 groups, respectively (Fig. 4C). There were 19, 299, 55, and 187 different volatile metabolites in the four groups of samples before and after fermentation, respectively (Fig. 4D). In order to screen the key differential metabolites of CTL before and after fermentation and further analyze the relationship between the four groups of differential metabolites, the 12 different non-volatile metabolites that DH, LC, PE, and YX share were "methoxyindoleacetic acid," "γ-glutamylphenylalanine," "N-benzoyl-2-aminoethyl-β-D-glucopyranoside," "L-tryptophan, 4-methyl-5-thiazoleethanol," "menatetrenone (vitamin K2)," "LysoPE 18:1 (2n isomer)," "3-pyridine-methanol-O-β-D-glucopyranosyl," "N′,N″,N‴-p-coumaroyl-cinnamoyl-caffeoyl spermidine," "LysoPC 14:0," "solanidine," and "γ-glutamyltyrosine" (Fig. 4E). The volatile metabolites shared by the four groups of CTL before and after fermentation were "carotol," "1,4-methanoazulen-9-ol, decahydro-1,5,5,8a-tetramethyl," and "phenol, 4-(1,1,3,3-tetramethylbutyl)."

## Potential metabolic pathway of the key flavor compounds of fermented CTL

In the process of CTL solid fermentation, bacteria and fungi utilize the carbohydrates and nitrogenous compounds of tobacco leaves as carbon and nitrogen sources for reproduction and growth, respectively. The proteins were degraded into a variety of amino acids, which were further deaminated to form α-ketoacid and produce $NH_3$, which explains the very distinct and pungent ammonia smell experienced by the workers at the fermentation prophase. The starch, sugar, and pectin would first be oxidized and decomposed into glucose and further through a variety of ways, such as glycolysis, to produce $CO_2$ and $H_2O$, and release energy (Fig. 5), which also explains that the temperature in the pile core can rise to 45–55°C after stacking.

The differential metabolites before and after fermentation in four production areas were significantly enriched in the synthesis and metabolism of various amino acids, such as alanine, aspartate, glutamate, arginine, phenylalanine, tyrosine, tryptophan, beta alanine, valine, leucine, and isoleucine (Fig. 4; Table S6). The metabolic pathways significantly enriched in the comparison before and after fermentation are also related to "nitrogen metabolism," "biosynthesis of various alkaloids," "carbon metabolism,"

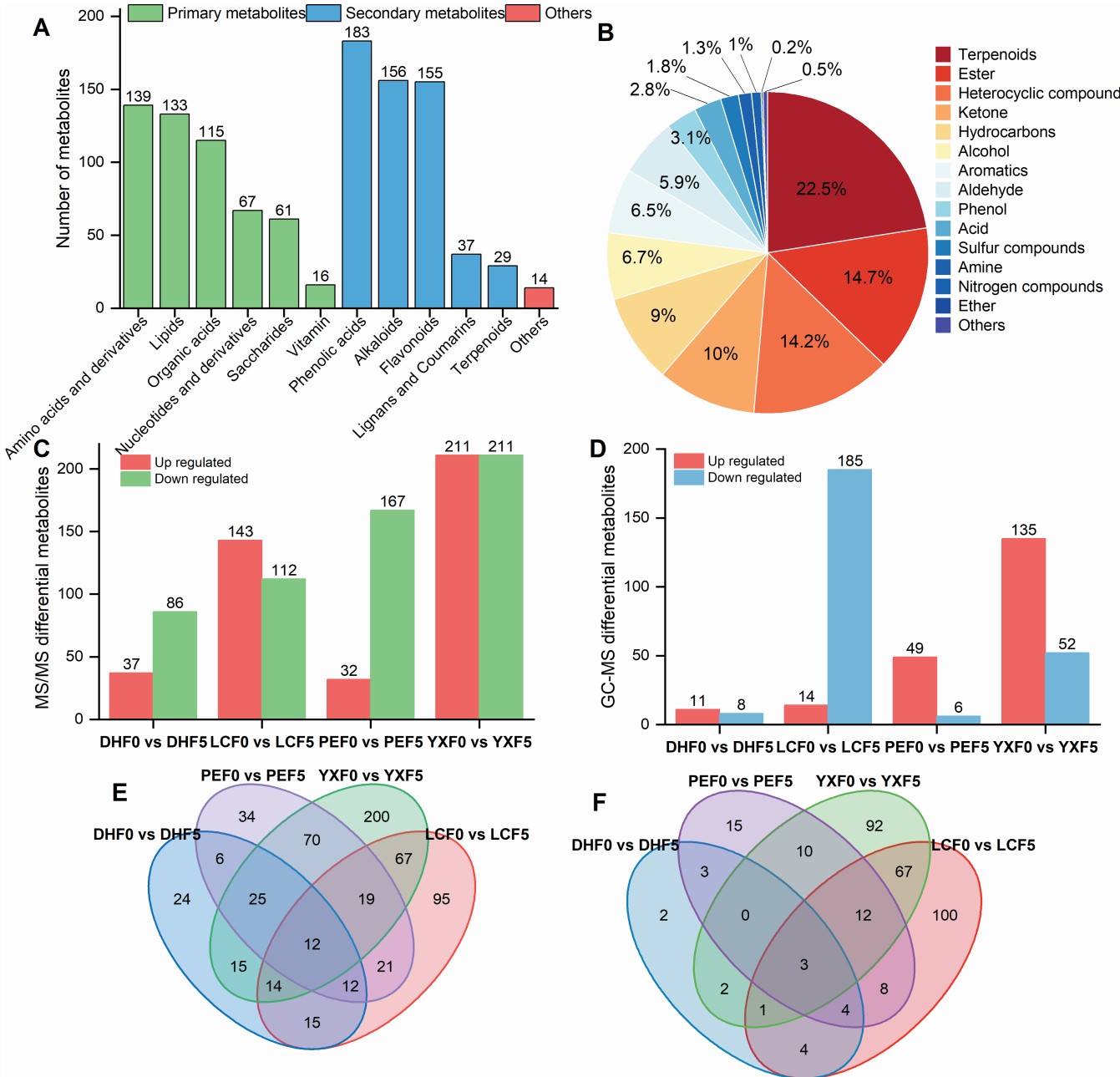

**FIG 4** Comparative analysis of non-volatile and volatile metabolites before and after fermentation. (A) Types and classification of non-volatile metabolites in CTL; (B) types and proportion of volatile metabolites in CTL; (C) non-volatile differential metabolites before and after fermentation from four origins; (D) volatile differential metabolites before and after fermentation from four origins; (E) Venn diagram shows specific and shared non-volatile differential metabolites; and (F) Venn diagram shows specific and shared volatile differential metabolites.

"flavonoid biosynthesis," "glycolysis," "pentose phosphate pathway," and "galactose metabolism" (Fig. 5). The pathway of carbon and nitrogen metabolisms shows that most carbohydrates were downregulated, and most nitrogen-containing secondary metabolites were upregulated. We hypothesized that the carbon–nitrogen coupled metabolic pathway promoted the synthesis efficiency of volatile flavor compounds, such as nitrogen-containing compounds (pyridine, pyrazine), heterocyclic compounds, terpenes, and aromatic compounds, in the CTL stacking fermentation system.

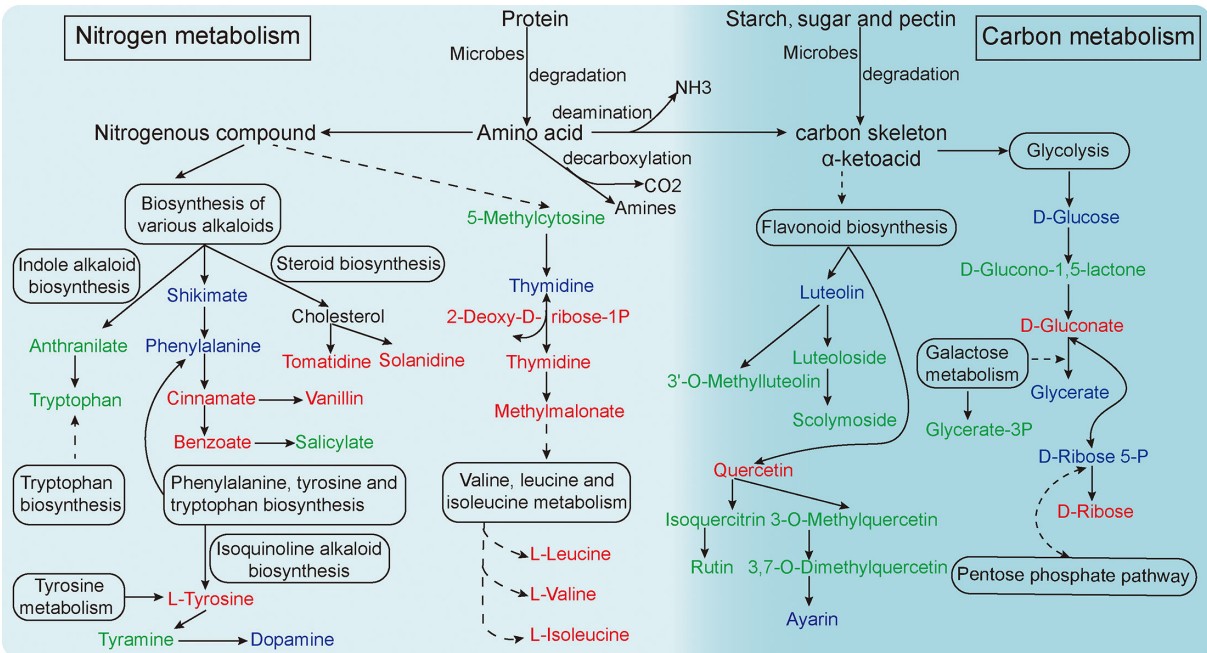

**FIG 5** Potential flavor metabolic pathways of macromolecules during CTL fermentation. The red means the metabolite content is significantly upregulated (F0 vs. F5); green means significantly downregulated; and blue means no significant change.

## Relationships between bacterial communities and volatile metabolites

In this study, the dominant bacteria with the top 50 relative abundances at the genus level were selected to analyze the correlations to volatile metabolites using the Spearman correlation coefficient. For DHF0 vs. DHF5, *Pseudomonas*, *Corynebacterium*, and *Massilia* showed negative or positive correlation with the other 19 volatile metabolites, among which XMW1182 and XMW0690 had the strongest correlation ($P < 0.01$) (Fig. 6A; Table S4). For LCF0 vs. LCF5, the selected bacteria were divided into Cyanobacteria, Proteobacteria, and Firmicutes by cluster analysis. XMW0775, KMW0482, KMW0456, XMW1182, NMW0113, KMW0430, XMW0164, KMW0528, KMW0530, WMW0010, XMW1408, WMW0021, XMW1099, KMW0574, KMW0102, NMW0766, XMW0600, and XMW0126 were most associated with nine core bacteria ($|r| >= 1$, $P = 0$) (Fig. 6B; Table S4). For PEF0 vs PEF5, a total of 27 bacteria were significantly associated with volatile metabolites, each from eight phyla. XMW0555, XMW0423, NMW0308, XMW0940, XMW0558, and NMW0256 were significantly negatively correlated with core bacteria ($r >= -1$, $P = 0$) (Fig. 6C; Table S4). For YXF0 vs. YXF5, 11 bacteria from four phyla were significantly associated with volatile metabolites, and 48 volatile metabolites were completely correlated with these dominant bacterial communities ($|r| >= 1$, $P = 0$) (Fig. 6D; Table S4).

## Relationships between fungal communities and volatile metabolites

This study further analyzed the relationship between dominant fungal communities (top 50) and volatile metabolites. The results showed that for DHF0 vs. DHF5, five fungi were associated with volatile compounds, and XMW0659, XMW0228, XMW0076, and KMW0345 were strictly linear with *Inocybe*, *Aspergillus*, and *Ascomycota*, respectively ($|r| = 1$, $P = 0$). (Fig. 7A; Table S5). For LCF0 and LCF5, 13 different fungi were associated with volatile compounds, among which WMW0203, XMW0563, XMW0010, KMW0213, and XMW0946 had a complete linear relationship with *Ascomycota* and *Candida* ($|r| = 1$, $P = 0$). (Fig. 7B; Table S5). For PEF0 and PEF5, the correlation heatmaps showed that 12 different fungi were associated with volatile compounds, among them, XMW1486,

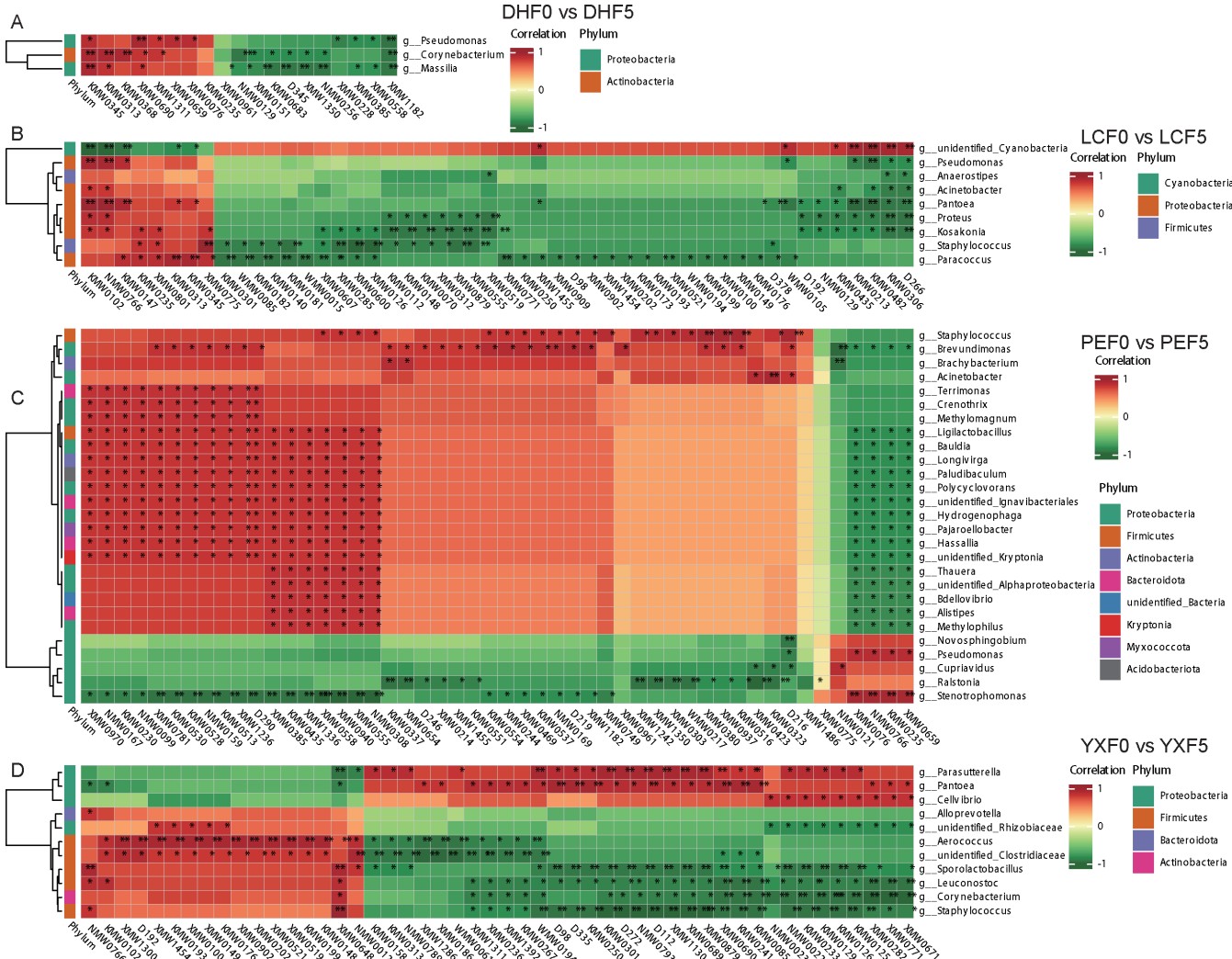

**FIG 6** Heatmap of relationships between volatile compounds and dominant bacteria. The microbes were clustered by UPGMA based on Euclidean distance. Hierarchical clustering heatmap between bacteria and volatile compounds from DHF0 vs DHF5 (A), LCF0 vs LCF5 (B), PEF0 vs PEF5 (C), and YXF0 vs YXF5 (D).

KMW0323, KMW0435, NMW0121, XMW1336, and XMW0385 have strict linear relationships with *Sampaiozyma*, *Microascus*, *Penicillium*, and *Aspergillaceae*, respectively (|r| = 1, *P* = 0) (Fig. 7C; Table S5). For YXF0 and YXF5, correlation heatmaps showed that 15 different fungi were associated with volatile metabolites, of which 41 volatile compounds had absolute linear relationships with nine fungi (|r| = 1, *P* = 0). They were *Aspergillus*, *Inocybe*, *Rhizopus*, *Cercospora*, *Colletotrichum*, *Aspergillaceae*, *Penicillium*, *Diaporthe*, *Microascus*, and *Aspergillus* (Fig. 7D; Table S5).

## DISCUSSION

CTL fermentation is a complex process involving many microorganisms and produces many metabolites. The growth and interaction of microorganisms play an important role in the unique flavor development of CTL (1, 2). 16S rDNA sequencing revealed that there were some differences in the core bacteria during CTL fermentation in the four origins (Fig. 2C). These bacteria have functions related to protein families, carbohydrate, amino acid, energy metabolism, and metabolism of cofactors and vitamins (Fig. S2A). Overall, *Cyanobacteria*, *Pseudomonas*, and *Staphylococcus* were the most metabolically active bacteria in the fermentation process (Fig. 2D). The isolation, identification, and biological activity of microorganisms showed that the culturable microorganisms were mainly

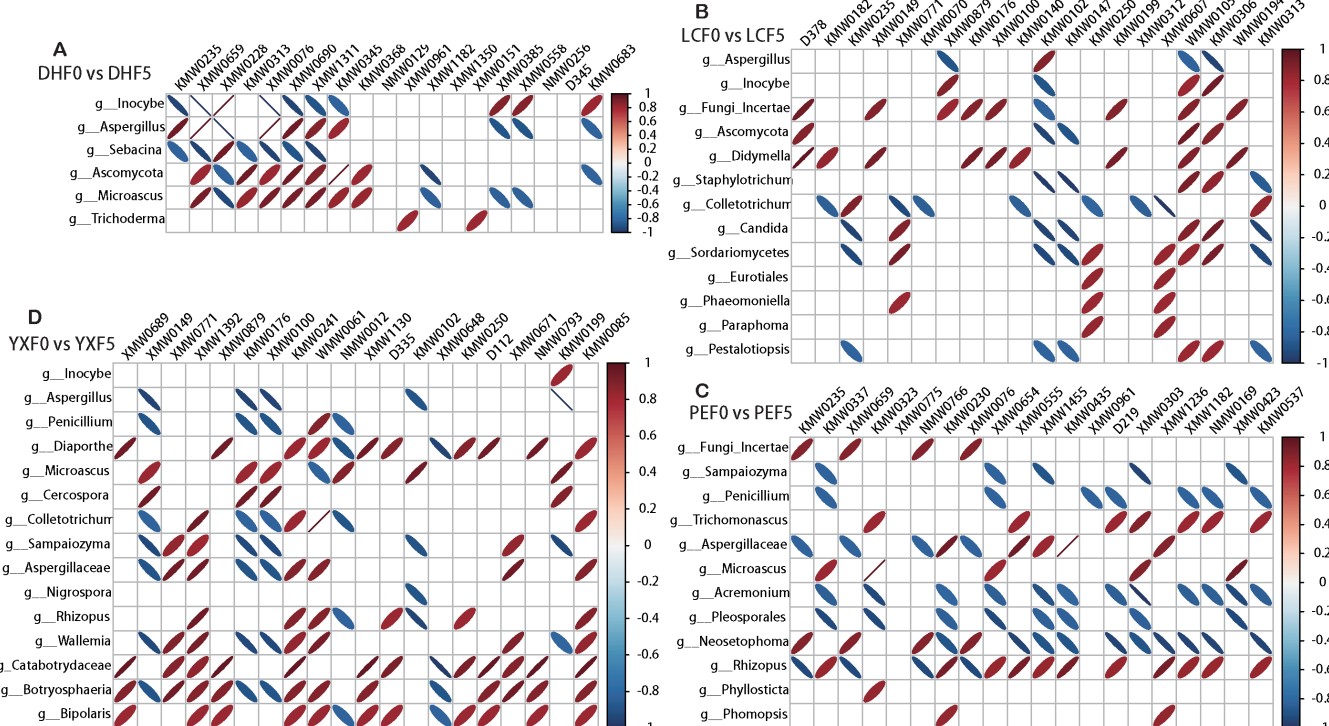

**FIG 7** Heatmap of relationships between volatile compounds and dominant fungi. Correlation heatmap between fungi and volatile compounds from DHF0 vs DHF5 (A), LCF0 vs LCF5 (B), PEF0 vs PEF5 (C), and YXF0 vs YXF5 (D). The red ovals represent a positive correlation, while the blue ovals represent a negative correlation. The greater the absolute value of the correlation, the finer the ellipse. The blank indicates $P > 0.01$.

*Bacillus*, *Pseudomonas*, *Alternaria*, and *Aspergillus* (Table S2), in which *Pseudomonas fulva* and *Aspergillus cristatus* could degrade protein, starch, lignin, and nicotine (Table 1; Fig. S1). Previous studies have also shown that *Pseudomonas* and *Staphylococcus* are the dominant bacteria in the fermentation process, and *Staphylococcus* plays an important role in improving the aroma of CTL (10). *Aspergillus cristatus* and *Pseudomonas fulva* are key species involved in the decomposition of starch, sugar, and other macromolecular constituents in the aging process of cigar (11). In addition, the correlation analysis of this study showed that *Staphylococcus*, *Cyanobacteria*, and *Corynebacterium* were absolutely correlated with 34, 15, and 8 volatile metabolites, respectively (Table S4), most of which belonged to terpenoids, ester, aromatics, ketone, acid, phenol, and amine substances.

In addition to transcriptional activity in bacteria, multiple filamentous fungi, including *Aspergillus*, *Penicillium*, and *Inocybe* fungal genera, showed the highest activity in CTL fermentation (Fig. 3D and E). FUNGuild functional prediction indicates that these core groups are symbiotroph and saprotroph, which obtain nutrients by exchanging resources with host cells and degrading dead host cells, respectively (Fig. S2B). *Aspergillus* and *Penicillium* were considered dominant fungi in the fermentation of various types of CTL (1, 12), participating in the degradation of macromolecular substances, such as starch and carbohydrates, in CTL (11), and had strong abilities to synthesize and secrete different enzymes (cellulase) and degrade proteins, starch, and cellulose to produce volatile flavors and organic acids (8). Furthermore, *Aspergillus* has a completely linear relationship with 15 volatile compounds, mainly terpenoids and ketones, while *Penicillium* is closely related to ester (Table S5). We found for the first time that *Inocybe* fungal genera have a large relative abundance during CTL fermentation and are closely related to the metabolism of terpenoids and aromatics. However, its function and effect on the flavor development of CTL fermentation need to be further verified. There are many kinds of microorganisms in CTL (13). Although most microorganisms are not culturable at present, it is very important to explore culturable microbial resources

for improving the fermentation process of CTL. Previous research has found that the predominant microorganisms were essential for the formation of key flavor qualities in CTL (14). In this study, the microbial and metabolome profiles and their relationships before and after CTL fermentation were systematically analyzed, and the relationships between six key functional microorganisms and metabolites were obtained, four of which were reported for the first time in CTL fermentation, and their degradation functions on macromolecular components were verified.

The synthesis and degradation metabolism of nitrogen-containing compounds, such as various amino acids and alkaloids, may be the key metabolic pathway for characteristic flavor development in CTL fermentation. Amino acids are important precursors of flavor substances, such as esters, heterocycles, pyridine, and pyrazines (5). Nitrogen-containing compounds, such as amines, pyridines, pyrazines, and nitriles, are volatile flavor components in CTL (15), which are mainly produced through various biosynthetic metabolic pathways of alkaloids. Our previous work also found that due to the multiple metabolic pathways involved in CTL during fermentation, flavonoids and lignans were significantly upregulated, while phenolic acids, amino acids, and derivatives were downregulated (4). The biosynthesis pathway of flavonoids is an active metabolic pathway during the fermentation process, and the biological transformation mechanism involves the molecular rearrangement of flavonoids. The carbon and nitrogen balance plays an important role in the growth and metabolism of microorganisms. Our previous studies found that the total sugar content decreased significantly after CTL fermentation, while the total nitrogen content did not change significantly (1), indicating that the consumption rate of carbon source was significantly higher than that of nitrogen source in the fermentation process.

In this study, amplicon sequencing was used to detect many microbial communities, but only a few culturable microorganisms were isolated and identified, and bacteria accounted for a relatively large proportion. Moreover, the types of microorganisms isolated from CTL before and after fermentation were different; for example, *Staphylococcus* was only isolated at the end of fermentation (Table S2), which was also consistent with sequencing results. Therefore, the study of microbial function may not only focus on the species and abundance of microorganisms but also on the culturability of microorganisms and the ability to express functional genes. Differences in metabolites in CTL may also change the degradation ability of microorganisms.

Based on microbial community succession and biodegradability, metabolic profile and pathway, metabolome, and metagenomic association analysis, we speculated that *Staphylococcus*, *Pseudomonas,* and *Cyanobacteria* mainly participated in the metabolic process of various amino acids and alkaloids, triggering phenylalanine, tyrosine, and tryptophan biosynthesis. As a result, cinnamate, vanillin, benzoate, and L-tyrosine were significantly increased (Fig. 2, Table 1, Fig. 5 and Table S5). *Aspergillus*, *Penicillium*, and *Inocybe* were mainly involved in the degradation of highly molecular compounds, such as polysaccharides, to form carbon skeletons, which further triggered the metabolic transformation of small molecular compounds in CTL. For example, the synthesis of quercetin was promoted, and the content of isoguercitrin and rutin was reduced after further degradation (Fig. 3, Table 1, Fig. 5 and Table S6). Finally, various amino acids and derivatives, alkaloid derivatives, organic acids, phenolic acids, free fatty acids, and other volatile flavor substances were formed. Together, these substances determine the typical flavor of CTL.

In summary, the balance of carbon and nitrogen should be controlled during CTL fermentation. It is necessary to comprehensively consider the carbon nitrogen ratio, temperature, moisture, pH value, and oxygen content and further improve the fermentation level and product quality by monitoring the fermentation process and optimizing the process parameters. The improvement of CTL quality is determined by the whole microecosystem during fermentation.

## MATERIALS AND METHODS

### Experimental materials

Cigar tobacco (*Nicotiana tabacum*, Yunxue No. 1) was grown in Lincang (LC), Puer (PE), Dehong (DH), and Yuxi (YX) in Yunnan. Central tobacco leaves of 1,000 kg were selected from three locations and sent to the fermentation factory in Yuxi for standardized artificially controlled fermentation. The fermentation experiments were carried out with pest control, sorting, rehumidification, balancing, stacking, fermenting, repiling (five times), and unstacking (Fig. 1). Repiling occurred immediately when the temperature reached 40–45℃. Approximately 500 g of tobacco leaves taken before fermentation (F0) and after fermentation (F5) was used in the following analyses. Three independent biological replicates were analyzed for each location.

### Microbial isolation, purification, and biological activity

The isolation and purification methods of microorganisms were referred to the reported methods. The single strains with strong growth were obtained by plate scribbling. The purified strains were attached to the solid medium containing starch, cellulose, nicotine, protein, and pectin and cultured at 37 for 24 h to determine the degradation ability of the strains. Degradation capacity = (hydrolytic ring diameter − colony diameter)/colony diameter.

### Microbial DNA extraction and sequencing

Total genomic DNA from the CTL samples and microbial strains was extracted using the cetyltrimethylammonium bromide method. The V4 regions of the bacterial 16S rRNA gene were amplified using the forward primer 515F (5′-GTGCCAGCMGCCGCGGTAA-3′) and the reverse primer 806R (5-GGACTACHVGGGTWTCTAAT-3′). The fungal internal transcribed spacer gene was amplified using the universal primers ITS1F (5-CTTGGTC ATTTAGAGGAAGTAA-3′) and ITS2R (5′-GCTGCGTTCTTCATCGATGC3′). The PCR product quantification and qualification, library preparation, and sequencing were carried out based on previous methods (16). The 250 bp paired-end reads were generated. The FLASH (Version 1.2.11) (17) was used to merge paired-end reads. Quality filtering was performed using the fastp software (Version 0.20.0) to obtain high-quality clean tags. The clean tags were compared with the reference database (Silva Database 138.1, https://www.arb-silva.de/ for 16S, Unite Database 2017.12, https://unite.ut.ee/ for ITS) using Vsearch (Version 2.15.0) (16) to detect and remove the chimera sequences. Denoise was performed with the DADA2 module in the QIIME2 software (Version QIIME2-202006) to obtain initial amplicon sequence variants, and then ASVs with abundances less than five were filtered out (18). Species annotation and multiple sequence alignment were performed using QIIME2 software. The absolute abundance of ASVs was normalized using a standard sequence number corresponding to the sample with the least sequences.

### HS-SPME–GC–GMS analysis

Chromatography and mass spectrometry were performed with the assistance of Wuhan Metware Biotechnology Co., Ltd. (Wuhan, China). Volatile metabolites in CTL were analyzed by HS-SPME–GC–MS. A total of 1.5 g CTL powder was placed in a 10 mL glass vial and extracted by headspace solid phase microextraction (120 μm DVB/CAR/PDMS fiber, Supelco, Bellefonte, USA) at 60℃ for 15 min. After extraction, volatile metabolites were identified according to those previously reported (19). MS was operated in selected ion monitoring mode for the identification and quantification of metabolites.

## UPLC–ESI–MS/MS analysis

The sample extracts were analyzed using an UPLC–ESI–MS/MS system (UPLC, ExionLC AD, https://sciex.com.cn/; MS, Applied Biosystems 6500 Q TRAP, https://sciex.com.cn/). The analytical conditions and the ESI source operation parameters were performed according to previously reported (4). A specific set of MRM transitions was monitored for each period according to the metabolites eluted within this period.

## Microbiome sequencing data analysis

The alpha and beta diversities of microbial communities were calculated with QIIME2 using weighted UniFrac distance between samples for bacterial 16S rRNA reads and Bray–Curtis dissimilarity for fungal ITS reads (20). Principal coordinate analysis was performed to evaluate the distribution patterns of microbiomes based on β-diversity calculated by the Bray–Curtis distance with the LabDSV R package. The specific method of Tax4Fun functional prediction is to cluster the ASV information of samples with the SILVA Database sequence as the reference sequence to obtain functional annotation information. A Python-based tool that can be used to taxonomically parse fungal ASVs by ecological guild, independent of sequencing platform or analysis pipeline (http://www.funguild.org/), was used. The sequencing results of single strains were compared with those in the National Center for Biotechnology Information (https://www.ncbi.nlm.nih.gov/) by BLAST to search for homologous sequences and identify the strain with the highest similarity to determine the biological classification.

## Metabolome data analysis

According to the self-built database MWDB (Metware Biotechnology Co., Ltd., Wuhan, China) and the public database of metabolite information, primary and secondary mass spectrometry data were used to conduct a qualitative analysis via referencing existing mass spectrometry databases. Unsupervised principal component analysis was performed using the statistics function prcomp within R (www.r-project.org). The hierarchical cluster analysis (HCA) results of samples and metabolites were presented as heatmaps with dendrograms, while Pearson correlation coefficients (PCC) between samples were calculated using the cor function in R and presented as heatmaps. Both HCA and PCC were carried out using the R package ComplexHeatmap. The normalized signal intensities of metabolites (unit variance scaling) were visualized as a color spectrum for HCA analysis. For the two-group analysis, differential metabolites were determined by VIP (VIP $\geq$ 1) and absolute $Log_2FC$ ($|Log_2FC| \geq 1.0$). VIP values were extracted from the OPLS-DA result, which also contained score plots and permutation plots generated using the R package MetaboAnalystR. To avoid overfitting, a permutation test (200 permutations) was performed. Identified metabolites were annotated, and then mapped to the Kyoto Encyclopedia of Genes and Genomes (KEGG) pathway using the KEGG Database (http://www.kegg.jp/kegg/), and their significance was determined by the hypergeometric test's $P$-values.

## Metabolome and metagenomic association analysis

Spearman correlation, CCA, and Cytoscape were used to explore the co-occurrence and interaction patterns between differential core taxa and volatile metabolites using the Metware Cloud, a free online platform for data analysis (https://cloud.metware.cn). The criteria for the significant correlation between differential microorganisms and metabolites were correlation coefficient $|r| >= 0.9$ and $P < 0.01$ for the significance test.

## Statistical analysis and visualization

One-way analysis of variance was used to analyze the significant impact of different fermentation stages (F0 and F5) on the α- and β-diversities of microbial communities in CTL. The Metastats method was used to screen species with significant differences

between groups. The probability of $P < 0.05$ indicated that the differences were significant. The Venn diagrams, pie charts, stacking histograms, and other skeleton diagrams were executed using OriginPro 2023 (10.0.0.154) (https://www.originlab.com/).

## ACKNOWLEDGMENTS

This work was supported by the Project of Yunnan Daguan Laboratory through grant number YNDG202402XJ01 and China Tobacco Monopoly Bureau Grants through grant number 110202103018/2022530000241002.

Conceptualization, G.K. and Guanghai Zhang; methodology, H.Y. and X.H.; software, Gaokun Zhao; validation, Z.L. and T.Z.; resources, Y.W. and Z.L; data curation, M.H.; writing—original draft preparation, Guanghai Zhang; writing—review and editing, G.K. and Y.H; visualization, H.X.; and funding acquisition, G.K. All authors have read and agreed to the published version of the manuscript.

## AUTHOR AFFILIATIONS

[1]Yunnan Academy of Tobacco Agricultural Sciences, Kunming, Yunnan, China
[2]Yunnan Tobacco Company of China National Tobacco Corporation, Yunnan, China
[3]Yunnan Oriental Tobacco Co., Ltd., Baoshan, Yunnan, China
[4]Puer Branch of Yunnan Provincial Tobacco Company, Puer, Yunnan, China

## AUTHOR ORCIDs

Guanghai Zhang  http://orcid.org/0000-0002-1326-8570
Guanghui Kong  http://orcid.org/0009-0003-2326-3284

## AUTHOR CONTRIBUTIONS

Yue He, Project administration | Wanlong Yang, Methodology | Ziyi Liu, Investigation | Zhonglong Lin, Investigation | Tikun Zhang, Investigation | Xiaohui He, Data curation, Investigation | Huachan Xia, Data curation, Investigation, Resources, Software, Validation | Mengting Huo, Resources, Software, Validation.

## DATA AVAILABILITY

The raw sequencing data have been uploaded to the National Center for Biotechnology Information (NCBI) Sequence Read Archive (SRA) database under BioProject number PRJNA856456.

## ADDITIONAL FILES

The following material is available online.

Supplemental Material

**Fig. S1 (Spectrum01029-24-S0001.tif).** Degradation circle size of culturable bacteria from cigar tobacco leaves.
**Fig. S2 (Spectrum01029-24-S0002.tif).** Predicted functional profile of bacteria and fungi.
**Supplemental material (Spectrum01029-24-S0003.docx).** Legends for supplemental tables and figures.
**Table S1 (Spectrum01029-24-S0004.xlsx).** Quality control of high-throughput microbial amplicon sequencing.
**Table S2 (Spectrum01029-24-S0005.xlsx).** Classification of culturable bacteria and fungi in cigar tobacco leaves.
**Table S3 (Spectrum01029-24-S0006.xlsx).** Enriched KEGG metabolic pathway of non-volatile metabolites.
**Table S4 (Spectrum01029-24-S0007.xlsx).** Significant differential metabolite summary of HS-SPME–GC–MS and UPLC–ESI–MS/MS.

**Table S5 (Spectrum01029-24-S0008.xlsx).** Spearman correlation of 16S bacteria genus and volatile metabolites.

**Table S6 (Spectrum01029-24-S0009.xlsx).** Spearman correlation of ITS fungi genus and volatile metabolites.

## Open Peer Review

**PEER REVIEW HISTORY (review-history.pdf).** Accounting of the reviewer comments and feedback.

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
