## [Reviewer comments · Microbiology Spectrum]

Microbiology Spectrum

Integrated microbiology and metabolomics analysis reveal the fermentation process and the flavor development in cigar tobacco leaf

Guanghai Zhang, Yue He, Wanlong Yang, Ziyi Liu, Zhonglong Lin, Tikun Zhang, Xiaohui He, Huachang Xia, Mengting Huo, Heng Yao, Gaokun Zhao, Yuping Wu, and Guanghui Kong

Corresponding Author(s): Guanghui Kong, Yunnan Academy of Tobacco Agricultural Sciences

Review Timeline:

Submission Date:	April 23, 2024
Editorial Decision:	January 7, 2025
Revision Received:	January 15, 2025
Editorial Decision:	February 12, 2025
Revision Received:	February 27, 2025
Accepted:	March 25, 2025

Editor: Lorena Ruiz

Reviewer(s): Disclosure of reviewer identity is with reference to reviewer comments included in decision letter(s). The following individuals involved in review of your submission have agreed to reveal their identity: Liwei Hu (Reviewer #1); Qin Liu (Reviewer #2)

Transaction Report:

DOI: <https://doi.org/10.1128/spectrum.01029-24>

Re: Spectrum01029-24 (Integrated microbiology and metabolomics analysis reveal the fermentation process and the flavor development in cigar tobacco leaf)

Dear Prof. Guanghui Kong:

Thank you for the privilege of reviewing your work. Below you will find my comments, instructions from the Spectrum editorial office, and the reviewer comments.

While the revision has found merit in your research, which is presented in a clear and logical way, some concerns have arisen regarding its novelty as compared to other research published related to cigar leaf fermentation. Besides, the fact that all research reported in your work is exclusively based on omics analyses, without any experimental validation of any of the observations reported, limits the impact of the investigation and raise doubts on the suitability of the article for the journal. If the authors can further elaborate their work to convincingly address these two key points to improve their work, in addition to address those issues highlighted by the reviewer below, I'll be happy to consider publication of the modified version of the manuscript.

Revision Guidelines

Sincerely,
Lorena Ruiz
Editor
Microbiology Spectrum

Reviewer #1 (Comments for the Author):

In this paper, the authors investigated how the microbiota affects the 17 metabolic state and characteristic flavor development of cigar tobacco leaves in the 18 fermentation process through microbial metabolism and co-metabolism with the host. The experiments were well arranged and the logic is clear, the result has a certain novelty. The following are the some suggestion for revision:

1. For the metabolomics analysis result in the abstract, it is better to mention more several compounds with high content and large variation.
2. In discussion, it needs to compare to the results of previous studies, and what new discoveries this study has made on the change of metabolomics and microbiology analysis in the cigar fermentation process.
3. The results showed that core bacterial communities in the fermentation process were Cyanobacteria, Pseudomonas and Staphylococcus, and Aspergillus, Penicillium and Inocybe were the most metabolically active fungi. It is recommended that references be cited in the discussion to discuss the functions of these microorganisms.
4. A table needs to be added to the material section to clearly indicate the sampling place and sampling time of all experimental samples.

Dear Reviewers,

On behalf of my co-authors, we thank you very much for giving us an opportunity to revise our manuscript, we appreciate reviewers very much for their positive and constructive comments and suggestions on our manuscript. We have studied comments carefully and have made correction which we hope meet with approval. Revised portion are marked in red in the paper. The main corrections in the paper and the responds to the Reviewer's comments are as flowing:

Response to Reviewers

Reviewer#1

in this paper, the authors investigated how the microbiota affects the 17 metabolic state and characteristic flavor development of cigar tobacco leaves in the 18 fermentation process through microbial metabolism and co-metabolism with the host. The experiments were well arranged and the logic is clear, the result has a certain novelty. The following are the some suggestion for revision:

1. For the metabolomics analysis result in the abstract, it is better to mention more several compounds with high content and large variation.

Re: We have clearly pointed out in the abstract that the contents of menalterone, solanidine, γ -glutamylphenylalanine, carotenol, phenol, 4- (1,1,3, 3-tetramethylbutyl) were significantly different before and after fermentation.

2. In discussion, it needs to compare to the results of previous studies, and what new discoveries this study has made on the change of metabolomics and microbiology analysis in the cigar fermentation process.

Re: In the discussion, we compared the results of this study with those of previous studies and found that the microbial and metabolome profiles and their relationships before and after CTLs fermentation were systematically analyzed, and the relationships between six key functional microorganisms and metabolites were obtained, four of which were reported for the first time in CTLs fermentation and their degradation functions on macromolecular components were verified.

3. The results showed that core bacterial communities in the fermentation process were Cyanobacteria, Pseudomonas and Staphylococcus, and Aspergillus, Penicillium and Inocybe were the most metabolically active fungi. It is recommended that references be cited in the discussion to discuss the functions of these microorganisms.

Re: In the discussion, we focused on the function of Pseudomonas fulva and Aspergillus cristatus in combination with the newly added verification results on the biological activity of culturable microorganisms.

4. A table needs to be added to the material section to clearly indicate the sampling place and sampling time of all experimental samples.

Re: The flow chart of fermentation treatment, collection and analysis of test samples is added to the material section, and the sampling location and sampling time of all experimental samples are clearly marked.

Re: Spectrum01029-24R1 (Integrated microbiology and metabolomics analysis reveal the fermentation process and the flavor development in cigar tobacco leaf)

Dear Prof. Guanghui Kong:

Thank you for the privilege of reviewing your work. Unfortunately reviewers have found some more issues that need to be addressed before the manuscript can be accepted for publication. Below you will find my comments, instructions from the Spectrum editorial office, and the reviewer comments.

Revision Guidelines

Sincerely,
Lorena Ruiz
Editor
Microbiology Spectrum

Reviewer #1 (Comments for the Author):

1. Abstract line 25-26, "Menatetrenone, Solanidine, γ -Glutamylphenylalanine, Carotol, and Phenol", the first letter of each chemical should be in lower case .
2. line 83, ob-tained, "-" should be deleted here.
3. line 86, 113, Operational Taxonomic Units (OTU), the first letter of each word should be in lower case.
4. line 144, "OUT sequence of CTL samples", what is "OUT" here?

5. "degrade macromolecular substances such as protein, starch, lignin and nicotine", nicotine should be deleted because it is not macromolecular substance.

Reviewer #2 (Comments for the Author):

In this study, authors illustrated how cigar tobacco leaves (CTL) were influenced by core bacterial communities during the period of fermentation using metagenomics and metabolomics techniques. Authors found that CTL fermentation is a carbon-nitrogen coupling metabolism mediated by microbiota for the first time, and further identified 4 microorganisms that could degrade macromolecular substances which may contribute to the typical flavors of CTL.

However, there are still some questions needed to be answered by authors:

1. In 3.6 and 3.7, please further illustrate the reason of choosing bacteria, how these bacteria were ranked to 'top 50'?

And why authors just analyzed volatile metabolites? Given the fact that non-volatile metabolites were dominated.

2. Based on the quantified data of metabolites and figure 5, please further discuss which microorganism may involve in the specific route of carbon-nitrogen metabolism?

And how these microorganisms may contribute to the typical flavors of CTL?

please add at least one paragraph to discuss it deeply.

3. Please check the typo error, for example Line 10 'Baoshan'.

Dear Reviewers,

On behalf of my co-authors, we thank you very much for giving us an opportunity to revise our manuscript, we appreciate reviewers very much for their positive and constructive comments and suggestions on our manuscript. We have studied comments carefully and have made correction which we hope meet with approval. Revised portion are marked in red in the paper. The main corrections in the paper and the responds to the Reviewer's comments are as flowing:

Response to Reviewers

Reviewer #1:

1. Abstract line 25-26, "Menatetrenone, Solanidine, γ -Glutamylphenylalanine, Carotol, and Phenol", the first letter of each chemical should be in lower case .

Re: We have corrected the names of each compound in the manuscript.

2. line 83, ob-tained, '-' should be deleted here.

Re: We have corrected all similar errors in the entire manuscript.

3. line 86, 113, Operational Taxonomic Units (OTU), the first letter of each word should be in lower case.

Re: We have corrected all similar errors in the entire manuscript.

4. line 144, "OUT sequence of CTL samples", what is "OUT" here?

Re: This is a typo and we have changed it to OTU.

5. "degrade macromolecular substances such as protein, starch, lignin and nicotine", nicotine should be deleted because it is not macromolecular substance.

Re: We have corrected all similar errors in the manuscript.

Reviewer #2:

In this study, authors illustrated how cigar tobacco leaves (CTL) were influenced by core bacterial communities during the period of fermentation using metagenomics and metabolomics techniques. Authors found that CTL fermentation is a carbon-nitrogen coupling metabolism mediated by microbiota for the first time, and further identified

4 microorganisms that could degrade macro-molecular substances which may contribute to the typical flavors of CTL.

However, there are still some questions needed to be answered by authors:

1. In 3.6 and 3.7, please further illustrate the reason of choosing bacteria, how these bacteria were ranked to 'top 50'?

And why authors just analyzed volatile metabolites? Given the fact that non-volatile metabolites were dominated.

Re: In this study, the dominant bacterial fungi with relative abundance in the top 50 genera were selected and their correlation with volatile metabolites was analyzed. Other food fermentation studies have shown that the relative abundance of microorganisms is positively correlated with their effects. Therefore, the authors believe that microorganisms with low abundance play a relatively small role in CTL fermentation.

The development process of typical flavor of CTL fermentation was studied in this paper. Volatile small molecules are the main contributors of flavor. Therefore, this study only analyzed the association between volatile metabolites and microbial communities.

2. Based on the quantified data of metabolites and figure 5, please further discuss which microorganism may involve in the specific route of carbon-nitrogen metabolism?

And how these microorganisms may contribute to the typical flavors of CTL? please add at least one paragraph to discuss it deeply.

Re: We discuss the metabolic pathways that microorganisms may be involved in and how they affect the typical flavor of CTLs. The details are as follows: “Based on microbial community succession and biodegradability, metabolic profile and pathway, metabolome and metagenomic association analysis. We speculated that *Staphylococcus*, *Pseudomonas* and *Cyanobacteria* mainly participated in the metabolic process of various amino acids and alkaloids, triggering phenylalanine, tyrosine and tryptophan biosynthesis. As a result, cinnamate, vanillin, benzoate and L-tyrosine were significantly increased (Figure 2, Table 1, Figure 5, Table S5). The *Aspergillus*, *Penicillium* and *Inocybe* were mainly involved in the degradation of high

molecular compounds such as polysaccharides to form carbon skeleton, which further triggered the metabolic transformation of small molecular compounds in CTL. For example, the synthesis of quercetin was promoted, and the content of isoguercitrin and rutin was reduced after further degradation (Figure 3, Table 1, Figure 5, Table S6). Finally, various amino acids and derivatives, alkaloid derivatives, organic acids, phenolic acids, free fatty acids and other volatile flavor substances were formed. Together, these substances determine the typical flavor of CTL.”

3. Please check the typo error, for example Line 10 'Baoshan'.

Re: We have corrected all similar errors in the manuscript.

Re: Spectrum01029-24R2 (Integrated microbiology and metabolomics analysis reveal the fermentation process and the flavor development in cigar tobacco leaf)

Dear Prof. Guanghui Kong:

Your manuscript has been accepted, and I am forwarding it to the ASM production staff for publication. Your paper will first be checked to make sure all elements meet the technical requirements. ASM staff will contact you if anything needs to be revised before copyediting and production can begin. Otherwise, you will be notified when your proofs are ready to be viewed.

Sincerely,
Lorena Ruiz
Editor
Microbiology Spectrum